# Diversity and Network Relationship Construction of Soil Fungal Communities in *Lactarius hatsudake* Tanaka Orchard during Harvest

**DOI:** 10.3390/microorganisms11092279

**Published:** 2023-09-11

**Authors:** Airong Shen, Baoming Shen, Lina Liu, Yun Tan, Liangbin Zeng, Zhuming Tan, Jilie Li

**Affiliations:** 1Institute of Forest and Grass Cultivation, Hunan Academy of Forestry, Changsha 410004, China; shyy123@hnlky.cn (A.S.); shenbm@hnlky.cn (B.S.); omphalina@outlook.com (L.L.); tany@hnlky.cn (Y.T.); 2Institute of Bast Fiber Crops, Chinese Academy of Agricultural Sciences, Changsha 410205, China; 3Hunan Provincial Key Laboratory of Forestry Biotechnology, Central South University of Forestry and Technology, Changsha 410004, China; lijilie12@163.com

**Keywords:** *Lactarius hatsudake*, soil microbiota, plantation, ecological network

## Abstract

*Lactarius hatsudake* Tanaka is a mycorrhizal edible mushroom with rich economic and nutritional value. Although it is artificially planted, its yield is unstable. Soil fungi, including *L. hatsudake*, coexist with many other microorganisms and plants. Therefore, complex microbial communities have an influence on the fruiting body formation of *L. hatsudake*. *L. hatsudake* and its interactions with the rest of the fungal community over time are not completely understood. In this study, we performed high-throughput sequencing of microorganisms in the basal soil of the fruiting body (JT), mycorrhizosphere soil (JG), and non-mushroom-producing soil (CK) in a 6-year-old *L. hatsudake* plantation at harvest. The results showed that the soil of the *L. hatsudake* plantation was rich in fungal communities and a total of 10 phyla, 19 classes, 53 orders, 90 families, 139 genera, and 149 species of fungi were detected. At the phylum level, the major groups were Basidiomycota and Ascomycota. At the genus level, the dominant groups were *Lactarius*, *Trichoderma*, *Suillus*, and *Penicillium*. Among them, *L. hatsudake* had an absolute dominant position in the soil fungal community of the plantation, and was the only group of *Lactarius* in the plantation soil. *Penicillium cryptum* and *Penicillium adametzii* were unique to the JT soil sample. *Chaetopsphaeria*, *Myxocephala*, *Devriesia*, and *Psathyrella* were positively correlated with *L. hatsudake*. In the constructed fungal network, the total number of nodes were ranked in descending order as JG (441) > CK (405) > JT (399), while the total number of edges were ranked in descending order as CK (1360) > JG (647) > JT (586). Analysis of the fungal assembly process revealed that groups CK and JG have determinative processes that dominated community building, while the JT group exhibited a dominant random process with a 0.60 probability. The results indicated that *L. hatsudake* was successfully colonized in the plantation soil. During harvest, the CK group exhibited the largest network size and the most complex fungal interactions, while the fungal community structure in the mushroom cultivation zone (JT and JG) was stable and less susceptible to external environmental interference. *L. hatsudake* affects the fungal community in the soil surrounding its fruiting body.

## 1. Introduction

*Lactarius hatsudake* belongs to the Basidiomycotina, Hymenomycetes, Agaricales, Russulaceae, and *Lactarius* fungi. It is a mycorrhizal edible mushroom that symbiotically grows with *Pinaceae* and *Quercus* plants [1,2]. The fruiting body of *L. hatsudake* is nutritious, delicious, and highly valuable as a medicinal and edible fungus. It is rich in bioactive compounds such as fungal polysaccharides, unsaturated fatty acids, nucleoside derivatives, various amino acids, and vitamins; it also possesses anti-tumor, anti-viral, cholesterol-lowering properties, as well as gut-health-promoting effects [3,4,5]. Thus, it is a perfect natural multi-functional food source. Moreover, *L. hatsudake* is a traditional delicacy in China and has various regional names, such as Hanjun, Yanejun, Congjun, Tongluojun, Simaojun, and Zihuajun. It has become one of the major commodities in the Chinese wild edible mushroom trade market and is also popular in Japan, Korea, Thailand, and other regions [1,6].

Soil microorganisms play important roles in mycorrhizal edible mushrooms, affecting their mycelial growth, mycorrhizal formation, and fruiting body development during their entire growth and development processes [7,8]. They also contribute to the yield, quality, flavor, and the formation of flavor substance and shiro [9,10,11,12,13,14,15,16]. For example, Kataoka et al. [17] and Yang et al. [7] found that soil microorganisms in the *Tricholoma matsutake* habitat can regulate the exchange of substances between the ectomycorrhizal fungi and host plants, and play a crucial role in mycelial growth and development as well as the fruiting body formation of *Tricholoma matsutake*. Napoli et al. [18] revealed the dynamic population of soil fungi to be related to the formation of *Tuber melanosporum* clusters. Miguel et al. [16] suggested that some fungi, such as species of *Boletus*, *Scleroderma*, *Pisolithus*, and *Trichophaea woolhopeia*, which often occur in truffle shiros, are closely related to the high yield of target truffles, while other fungi, such as the *Hebeloma*, *Laccaria*, and *Russula* species, are likely to reduce the production of target fruiting bodies. Furthermore, Francisco et al. [19] determined the yield of desert truffle *Terfezia claveryi* to be affected by the composition of soil and root fungus species, with the production in mycorrhizal roots positively correlated with ectomycorrhizal and arbuscular mycorrhizal fungi, and a positive correlation observed between fungi parasites and plant pathogens in non-producing roots [19]. To improve the yield and quality of mycorrhizal edible fungi with a high commercial value, researchers have analyzed the soil microbial community characteristics of mycorrhizal edible fungi, such as *Tricholoma matsutak*, and truffles to improve cultivation techniques and increase yield [8,16,19].

Currently, significant progress has been made in the establishment techniques of *L. hatsudake* plantations [20,21]. However, technical challenges such as the low production and unstable yield of fruiting bodies in plantations have not yet been resolved. This study selected a 6-year-old *L. hatsudake* plantation as the research object and employed high-throughput sequencing technology to study the fungal community structure, diversity, and ecological network of the basal soil of the fruiting body, mycorrhizosphere soil, and non-mushroom-producing soil during the harvest period. We also explored the impact of the artificial cultivation of *L. hatsudake* on the stability of the soil fungal community. The study aimed to: (i) identify the differences in the soil microbial community between the growth areas of *L. hatsudake* and the adjacent soil where the mushrooms do not grow; (ii) identify the dominant soil microbial populations closely related to the growth of *L. hatsudake*; and (iii) provide references for future research on the growth and development mechanisms, fine management of plantations, and rhizosphere soil microbiological functions of the *L. hatsudake* mycorrhizal roots and fruiting bodies in wild environments.

## 2. Materials and Methods

### 2.1. Plantation Overview

The sample collection site was located within an *L. hatsudake* plantation established by our team, situated in Pufeng Village, Puji Town, Liuyang City, Hunan Province, China (113°21′–113°31′ east longitude and 27°51′–28°02′ north latitude). The plantation with the host *Pinus massoniana* was established in the spring of 2014 with a row spacing of 3 m × 3 m. The first time of *L. hatsudake* production was autumn 2016. The climate is characterized by a subtropical monsoon with an even distribution of water and rain, a frost-free period of 271 days, 1656 h of sunshine, and a rainfall of 1552 mm per year [22]. The total study area was 10,005 m^2^, with an altitude of 126 m and west-facing slope. The dominant soil is red soil with a thickness of over 1 m.

### 2.2. Experimental Design and Soil Sample Collection

The sampling time for this study was 25 October 2020, which was the period in which the fruit bodies of *L. hatsudake* produced the most in a year. The plantation was divided into 25 duplicate samples of 20 m × 20 m. Eight non-adjacent samples were selected following a “Z”-shaped pattern, and within each selected sample, five mushroom-producing areas and five non-mushroom-producing areas were randomly selected for the sample collection. The mushroom-producing area refers to the area between the host tree’s canopy radiation scope and the base of the host tree, where 15 or more fruiting bodies are produced. The non-mushroom-producing area refers to the soil area without any host-tree canopy radiation and fruiting bodies are never produced. The distance between adjacent sampling points was about 2 m. The soil type of the plantation is red soil, with a pH value of 6.0–6.8 and a soil humidity of 45–50% when collecting samples.

The sampling followed the method described by Miae et al. [23]. Two sites of samples were collected for the mushroom-producing area: (i) The base soil (namely JT) was at the base of the *L. hatsudake* fruiting body. For the JT samples, fresh and healthy *L. hatsudake* fruiting bodies with a mushroom cap diameter of 3–6 cm and no insect damage were selected. Using sterile forceps, the fruiting body, including the soil within 5 cm of the base, was placed in a 250 mL sterile culture bottle, labeled, and transported back to the laboratory in an icebox. The soil at the base of each fruiting body was gently scraped off with a sterilized surgical blade on a clean bench. After mixing well, the samples were stored at −80 °C for future use. (ii) Mycorrhizosphere soil (namely JG). The *L. hatsudake-Pinus massoniana* mycorrhiza root system from the selected samples was excavated and placed into a sterilized culture bottle that was labeled and transported back to the laboratory in an icebox. The soil on the root system was brushed with a sterile brush, and after mixing well, the samples were stored at −80 °C for future use. Non-mushroom-producing area (CK) soil samples were collected at each sampling point (0–5 cm), mixed well, and stored at −80 °C for future use. Eight samples were collected for each sample type, resulting in twenty-four soil samples.

### 2.3. DNA Extraction, PCR Amplification, and High-Throughput Sequencing

Total soil microbial DNA was extracted using an MOBIO Power Silo DNA Isolation Kit (Anbiosci Tech Ltd., Shenzhen, China). The specific steps followed the instructions provided in the kit manual. The DNA content and purity were analyzed using a nucleic acid quantification instrument, and the integrity of the DNA samples was tested using agarose gel electrophoresis. Samples with an OD260/280 between 1.8 and 2.0 were stored at −80 °C in a refrigerator.

The ITS1 region of the fungi-specific 18S rRNA gene was amplified and sequenced using specific primers with barcodes ITS5-1737F: 5′-GGAAGTAAAAGTCGTAACAAGG-3′ and ITS2-2043R: 5′-GCTGCGTTCTTCATCGATGC-3′, respectively. The PCR products were detected via electrophoresis on a 2% agarose gel and then mixed in equal amounts. The mixed products were purified using 2% agarose gel electrophoresis with 1× TAE and the target bands were then cut and recovered using a GeneJET Gel Extraction Kit (Thermo Fisher Scientific Baltics UAB, Vilnius, Lithuania).

An Ion Plus Fragment Library Kit (Life Technologies, Carlsbad, CA, USA) was used to construct libraries. After the libraries were quantified and checked using Qubit 3.0 Fluorometer (Thermo Fisher Scientific, Shah Alam, Malaysia), they were subjected to a quality control test. Following this, Ion S5TMXL (Thermo Fisher Scientific, Braunschweig, Germany) was used for sequencing at Beijing Novogene Technologies Co., Ltd., Beijing, China.

### 2.4. Data Analysis

Cutadapt (v 1.9.1) was used to remove low-quality reads [24]. The reads were then split according to their barcodes to obtain the data for each sample. The barcode and primer sequences were removed, and the final effective data were obtained by eliminating the chimeric sequences. Chimeric sequences were detected by comparing the reads with the annotation databases using (https://github.com/torognes/vsearch/, accessed on 29 May 2023) [25,26].

#### 2.4.1. Community Diversity Analysis

The clean reads of all samples were clustered into operational taxonomic units (OTUs) with a similarity threshold of 97% using USEARCH (v10.0) [27]. The OTUs were filtered at a threshold of 0.005% of the total number of sequences for subsequent analyses. The Ribosomal Database Project (RDP) classifier [28] (with a confidence threshold of 0.8) was used to classify the OTUs based on the UNITE (fungi) taxonomy database [29]. The NMDS method in Mothur (v1.30) was used for multidimensional scaling analysis based on the OTUs [30]. Mothur was also employed to calculate various diversity indices, including the richness, Chao1, ACE, Shannon, and Simpson indices, based on the OTU results. A clustering heatmap of the soil microbial species was generated using the “heatmap” R package (v4.2.1). The beta diversity of the fungal community was analyzed using QIIME2 (v2020.6) [31] based on Bray–Curtis dissimilarity, and principal coordinates analysis (PCoA) and distance heatmaps were used to detect the differences in the fungal community structure.

#### 2.4.2. Analysis of Community Species Differences

LEfSe (http://huttenhower.sph.harvard.edu/galaxy, accessed on 29 May 2023) was employed to analyze the differential species between samples using the OTU abundance matrix. Differences were considered significant for linear discriminant analysis (LDA) scores greater than four and Kruskal–Wallis rank-sum test values less than 0.05.

#### 2.4.3. Molecular Ecological Network Construction and Analysis

After standardizing the OTU data obtained from high-throughput sequencing, a Spearman rank correlation matrix was constructed by uploading the data to the Molecular Ecological Network Analyses Pipeline (MENA) website. Based on random matrix theory (RMT) with default parameters, three molecular ecological networks of microbial communities from different plantation environments were constructed. The networks were further visualized and modularized using the interactive Gephi 0.9.7 platform [32]. The “induced_subgraph” function was employed to identify the topological network features of each soil sample. Based on the within-module connectivity (Zi) and between-module connectivity (Pi), all species were divided into four groups, namely, module hubs (Zi > 2.5), network hubs (Zi > 2.5 and Pi > 0.62), connectors (Pi > 0.62), and peripherals (Zi < 2.5 and Pi < 0.62). The species identified as module hubs, network hubs, or connectors were considered key species in the community.

#### 2.4.4. Soil Fungal-Community Assembly Process

The relative importance of the deterministic and stochastic processes in the community was evaluated using the Modified Stochasticity Ratio (MST), corrected by the R function “tNST”.

#### 2.4.5. Statistics and Analysis

The “aov” function in R was used to analyze the significant differences in the microbial diversity, richness indices, and species-composition structure. Data were visualized using the “ggplot2” R package.

## 3. Results

### 3.1. Overall Sequencing Results

A total of 24 samples collected from the *L. hatsudake* plantation were subjected to high-throughput sequencing using the Illumina Miseq platform, resulting in 1,700,813 clean reads after quality control and assembly. A total of 785 fungal OTUs were obtained from the 24 samples. Among the OTUs, 731 fungal OTUs were shared by the soil from the three different sites in the orchard. The CK site exhibited the most fungal OTUs (770) and 5 unique OTUs, followed by the JT site (763 OTUs) with 3 unique OTUs, and the JG site with the fewest fungal OTUs (759) and 1 unique OTU.

### 3.2. Analysis of Soil Fungal Diversity in the L. hatsudake Orchard during the Harvest Period

#### 3.2.1. Analysis of Fungal α-Diversity

There were no significant differences in richness, ACE, and other diversity indices between the fungal community data from the CK site and those from JG site (Table 1). However, these indices were significantly higher than those from the JT site, indicating that the α-diversity of the fungal community in the JT soil was lower than that in the CK and JG soils.

#### 3.2.2. Analysis of Fungal β-Diversity

The soil fungal communities in the three sites of the *L. hatsudake* orchard during the harvest period were relatively similar (Figure 1). The first plane formed by the two axes explained 63.07% of the differences between samples. The effects on the fungal community structure of the different samples varied between the two principal components, with the CK and JG sites exhibiting greater similarities and overlaps compared with the JT site. This indicates that there were fewer differences in the fungal diversity between these sites CK and JG. PCoA analysis among the three sites revealed that there were no significant differences between CK and JG, while significant differences were identified between each of these sites and JT. This is consistent with the conclusions of the α-diversity analysis.

### 3.3. Soil Fungal-Community Structure in the L. hatsudake Orchard during the Harvest Period

The species annotation detected a total of 10 phyla, 26 classes, 57 orders, 112 families, 157 genera, and 137 species of fungi. As shown in Figure 2, the three sites had similar phylum distributions. Basidiomycota and Ascomycota were the dominant soil fungi phyla in the *L. hatsudake* orchard during the harvest period, accounting for 98.02%, 94.47%, and 93.42% of the fungal communities at the JT, JG, and CK sites, respectively, with significant differences between these sites. In addition, there were six phyla with relative abundances greater than 0.01%, namely, Mortierellomycota, Glomeromycota, Mucoromycota, Chytridiomycota, Olpidiomycota, and Unclassified, among which Mortierellomycota, Chytridiomycota, and Mucoromycota showed significant differences.

Figure 3 presents the horizontal distribution bar chart and composition table of the main fungal communities revealed. Minimal differences were observed in the types of soil fungal communities between the mushroom-producing and non-mushroom-producing areas of the *L. hatsudake* orchard at the three sampling points during the harvest period. However, significant differences were identified in soil fungal-community content, with the *Lactarius* abundance value in the JT area reaching 68.60%, which was significantly higher than that in the JG and CK areas. Analysis of the top 30 genera by abundance revealed that there were 10 genera with an abundance exceeding 1.00% in the three sites, namely, *Lactarius*, *Trichoderma*, *Penicillium*, *Suillus*, *Talaromyces*, *Cladophialophora*, *Archaeorhizomyces*, *Tomentella*, *Amphinema*, and Unclassified. Furthermore, a difference of 16.67% was observed in the genera between the three areas. In particular, the abundance of *Lactarius* and *Penicillium* in the mushroom-producing areas (JT or JG) was identified as significantly higher than that in CK, while the content of *Trichoderma*, *Mortierella*, and Unclassified was significantly lower than that in CK. At the species level, the relative abundance of *L. hatsudake* and *T*. *avirens* in both the mushroom-producing and non-mushroom-producing areas exceeded 1%. The results identified *L. hatsudake* as the only species of the *Lactarius* genus in the orchard (Figure 3). The genus-level species-clustering heatmap demonstrated that the sequence evolutionary relationship of soil fungi in the JT group was close to that of the CK and JG groups (Figure 4).

Further research on the classification information of the genera and species in the two mushroom-producing areas identified several genera with significantly higher relative abundances in the JT and JG areas than in the CK area. These included *Penicillium* (2.95%, 2.56%), *Acremoniopsis* (0.01%, 0.08%), *Ophiocordyceps* (0.01%, 0.03%), *Fusidium* (0.06%, 0.02%), and *Gymnopus* (0.15%, 0.02%). These genera may have the potential to enhance the field growth and development of the *L. hatsudake* fruiting body and mycorrhiza. Unique genera in the JT area included *Lophiostoma* (0.01%), *Psathyrella* (0.06%), and *Mycosisymbrium* (0.02%), and may be related to the development and production of *L. hatsudake* fruiting bodies. Unique genera in the JG area included *Parengyodontium* (0.06%), *Didymella* (0.01%), *Ilyonectria* (0.01%), *Tausonia* (0.01%), *Myceliophthora* (0.01%), *Amorphotheca* (0.01%), *Symmetrospora* (0.01%), *Filobasidium* (0.01%), *Colletotrichum* (0.01%), *Metarhizium* (0.01%), and *Chaetomium* (0.01%). These genera may be beneficial in the formation and development of *L. hatsudake* mycorrhiza.

### 3.4. Analysis of the Differences in the Fungal Community of Different Sites in the L. hatsudake Orchard during the Harvesting Period

LEfSe analysis was performed based on the fungal community composition of different soil samples in the *L. hatsudake* orchard. Figure 5a,b depict the corresponding purifying branch graph and LDA value-distribution column chart, respectively. The analysis revealed the presence of significant biological indicator fungal species in the soil of different sites in the orchard during the harvesting period. The fungal biomarker species for the JT site included Russulales, *Lactarius*, *L. hatsudake*, Russulaceae, *Penicillium adametzii*, Agaricomycetes, and Basidiomycota (Figure 5b). These species may be related to the development of the *L. hatsudake* fruiting bodies. The biomarker fungi for the JG site included *Suillus*, Suillaceae, Atheliaceae, Athliales, and *Amphinema*. The biomarker fungi for the CK site included Ascomycota, *T. virens*, Hypocreales, *Trichoderma*, Hypocteaceae, Sordariomycetes, Rhizopogonaceae, *Podospora*, and *Rhizopogon*. This area had the most diverse fungal species among the three sites.

### 3.5. Molecular Ecological Network Structure Characteristics of Soil Fungi in the L. hatsudake Orchard during the Harvesting Period

Spearman rank correlation analysis was conducted to construct three molecular ecological networks based on high-throughput sequencing data (Table 2). The analysis identified 399, 441, and 405 OTU nodes for JT, JG, and CK, respectively, and described the overall network structure by calculating the main characteristic parameters. Highly similar thresholds of 0.89, 0.89, and 0.91 were determined for the three networks, respectively. The average path length and average clustering coefficient in the molecular ecological networks were higher than those in the random networks. The R2 values for the topological distributions of the three networks were 0.9, 0.802, and 0.775, respectively, which were consistent with the power-law. This indicates that the networks constructed in this study exhibited scale-free, small-world, and modular network characteristics that could be used for the subsequent study of fungal interrelationships.

The total node sizes in the fungal networks were observed, in descending order, as JG (441) > CK (405) > JT (399), and the total edge sizes, in descending order, as CK (1360) > JG (647) > JT (586). This indicates that during the harvesting period, the network of CK was the largest in terms of scale and had the most complex interrelationships between fungi, followed by JG and JT (Table 2). The network analysis results of the three sampling sites at different positions were consistent with the topological analysis results, providing us with a more intuitive display of the differences in the fungal networks of the three sites (Figure 6a). The fungal network size and complexity of the interrelationships between fungi in the mushroom-producing area of the plantation were smaller than those in the non-mushroom-producing area. This finding demonstrates that compared with the non-mushroom-producing area (CK), the fungal community structures in the production areas (JT and JG) were more stable and less susceptible to external environmental interferences (Table 2). The fungal network at the CK site generally exhibited positive correlations, with positive edges accounting for 57.9%. In contrast, the JT and JG sites mainly exhibited negative correlations, with negative edges accounting for 56.1% and 52.4%, respectively. This suggests that the competition relationship between the fungal species at the latter two sites (JG and JT) may be stronger compared with that at CK site.

The network diagram generated based on the soil fungal OTU profile included 108 nodes and 154 edges. Ascomycota, Basidiomycota, Mortierellomycota, and Glomeromycota were the dominant fungi in the nodes (Figure 6b). The network diagram contained eight modules, of which the main module accounted for 71.3% of the total nodes (Figure 6b). Moreover, 81.82% of the connections in the network were positive, indicating a strong level of cooperation between the soil fungi in the *L. hatsudake* orchard. The majority of the OTUs in the network were identified as peripherals, and only OTU181 (f-Lasiosphaeriaceae) and OTU185 (g-Podospora) were identified as module hubs. This reveals the crucial roles played by the family Lasiosphaeriaceae and the genus *Podospora* in maintaining the interdependence of the soil microbial community and regulating the microbial ecological community. We further investigated the correlation between *Lactarius* and other fungi, indicating that *Lactarius* was positively correlated with *Chaetopsphaeria*, *Myxocephala*, *Devriesia*, and *Psathyrella* and negatively correlated with *Amphinema*, *Rhizopogon*, *Tylospora*, *Coniosporium*, *Scleroderma*, *Arcopilus*, *Staphylotrichum*, *Hydropus*, *Geminibasidium*, *Mollisia*, *Clavaria*, *Podospora*, *Lauriomyces*, *Coniella*, *Thermoascus*, *Tolypocladium*, *Acephala*, *Hydropus*, and *Dendrochytridium* (Figure 6c).

### 3.6. Microbial-Community Assembly Process

Analysis of the MST revealed low values for the CK and JG groups, at 0.13 and 0.21, respectively, indicating that deterministic processes played a dominant role in the community assembly of these sites. The JT group exhibited a higher MST value, 0.60, which suggests that the stochastic process played a dominant role in the community assembly of this site (Figure 7).

## 4. Conclusions and Discussion

This article focuses on the soil fungal-community composition, diversity, and molecular ecological network of three sites, namely, the fungal production area of the *L. hatsudake* plantation (i.e., the base soil of the fruiting body, the mycorrhizosphere soil and the non-mushroom-producing soil). The soil in the *L. hatsudake* plantation area was found to have a rich and diverse fungal community, with 10 phyla, 19 classes, 53 orders, 90 families, 139 genera, and 149 species of fungi identified. The dominant taxa at the phylum level were Basidiomycota and Ascomycota, while the dominant genera at the genus level were *Lactarius*, *Trichoderma*, *Suillus*, and *Penicillium*. Notably, *L. hatsudake* had an absolute dominant position in the fungal community and was the only population of *Lactarius* in the plantation soil, indicating a successful establishment of this fungal species in the soil. This is distinct to the findings of Francisco et al., whereby *Chaetomium* was not among the top 10 genera identified in plantation soil. This difference may be due to the distinct survival strategies of different types of mycorrhizal edible fungi [19].

Multiple studies have determined the fungi of the *Penicillium* genus to be a dominant population in mycorrhizal edible fungus ponds such as matsutake, which may facilitate the growth and development of their mycorrhiza and fruiting bodies [3,6,33,34,35,36,37]. The current study also found that the relative abundance of *Penicillium* in the mushroom-production area (2.95% and 2.56% for JT and JG, respectively) was significantly higher than that in the non-mushroom-producing area of CK (0.79%). This is the dominant fungus in the *L. hatsudake* production area and one of the fungal biomarker species in JT. Interestingly, no correlation was observed between the *Penicillium* and *Lactarius* fungi in our soil fungal interaction network within the plantation. However, through differential analysis at the species level, we found that *Penicillium cryptum* and *Penicillium adametzii* were unique populations in JT. This phenomenon may require further research combining traditional cultivation methods and molecular methods. Mycorrhizal fungi such as *Suillus* and *Amphinema* were detected in the plantation. The number of mycorrhizal fungi species at the CK site was higher than that in the mushroom-production area (JT and JG). The relative abundance of some mycorrhizal fungi was highest in JG, which is in agreement with previous studies [16,38,39,40,41,42].

The relative abundance of *Chaetopsphaeria* and *Devriesia* in the mushroom-producing area (JT and JG) was significantly higher than that in the non-mushroom-producing area, and was positively correlated with *L. hatsudake*. At the JT site, *Lophiostoma*, *Psathyrella*, and *Mycosisymbrium* (which was positively correlated with *L. hatsudake*) were unique genera with lower relative abundances. The relative abundance of *Myxocephala* was the highest in JT, and was significantly higher than that in CK and JG. It was also positively correlated with *L. hatsudake*. These genera are likely to be related to the mycorrhiza and fruiting bodies of *L. hatsudake*. *Amphinema*, *Scleroderma*, and *Rhizopogon* were observed to have the highest relative abundances at the CK site, followed by the JG site, while JT exhibited the lowest relative abundances. The network correlation showed a negative correlation with *L. hatsudake*, indicating that these fungi may compete with *L. hatsudake*.

The shared and significantly high-abundance fungal genera in the two sites of JT and JG in the mushroom-producing area, which were higher than those in the non-mushroom-producing area, included *Luellia*, *Oidiodendron*, *Chaetosphaeria*, *Acremoniopsis*, *Ophiocordyceps*, *Fusidium*, and *Gymnopus*. These fungi may be beneficial for the fruiting body production and mycorrhizal development of *L. hatsudake*. The genus *Hydropus* was detected at the JG site with a relative abundance of 0.26%, which was significantly higher than that in the non-mushroom-producing area (CK). Note that *Hydropus* was not detected in JT. The genera *Parengyodontium*, *Didymella*, *Ilyonectria*, *Tausonia*, *Myceliophthora*, *Amorphotheca*, *Symmetrospora*, *Filobasidium*, *Colletotrichum*, *Metarhizium*, and *Chaetomium* were unique to the mushroom-producing area JG. The functional roles of these fungal genera in other types of mycorrhizal edible fungus plantations or natural habitats remain to be further explored and verified.

Alpha analysis indicated the fungal OTU number and Simpson index in the non-mushroom-producing area to be higher than those in the production area. This suggests that the non-mushroom-producing area had a higher species richness, diversity, and abundance of fungal species. Beta analysis showed that the fungal communities at the CK and JG sites overlapped to a certain extent, while those at the JT site were well-separated. At the phylum and genus levels, the taxa observed at the JG and JT sites were also generally found at the CK site. In the ecological network of the fungal community within the *L. hatsudake* plantation, the mushroom-producing sites (JT and JG) exhibited the smallest network size and simple interactions dominated by competition when compared with the non-mushroom-producing site CK. Negative connections of 56.1% and 52.4% were observed for JT and JG, respectively, while the positive connection in CK was 57.9%. Similar patterns have been observed in other edible ectomycorrhizal fungi, such as *Tuber aestivum*, *Tuber melanosporum*, *Tuber indicum*, and *Tricholoma matsutake*, in their natural habitats or plantations [23,43,44,45,46]. Liu et al. [47] found that the positive connection in the bare soil of the *Tuber indicum* plantation was higher than in the attached soil of the fruiting bodies. The authors showed that the bare soil contained a more complex fungal community, and the microbial diversity, evenness, and richness decreased from the soil to the attached soil, to the fruiting body surface, and finally to the fruiting body interior. Network analysis results showed that the networks of JG and JT were less complex compared with that of CK. Such a phenomenon may be attributed to the metabolites secreted by *L. hatsudake*, which inhibit the growth of other fungal species [14,18,35,48,49,50,51]. This may indicate that *L. hatsudake* can selectively enrich some soil fungi, similar to ecological filtering, to facilitate its growth and development. This may be attributed to the absolute dominant position of *L. hatsudake* at the JT site and may have affected the fungal community in the surrounding soil.

## Figures and Tables

**Figure 1 microorganisms-11-02279-f001:**
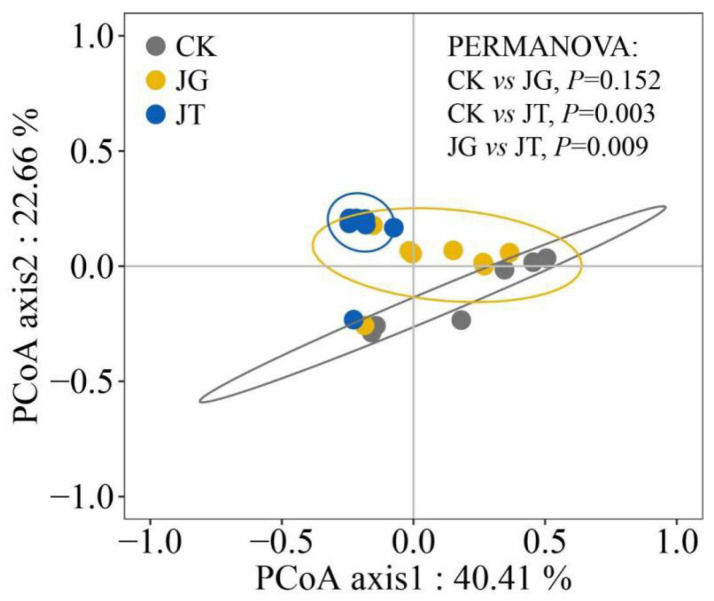
Principal coordinates analysis (PCoA) of soil fungal-community composition in the *L. hatsudake* orchard.

**Figure 2 microorganisms-11-02279-f002:**
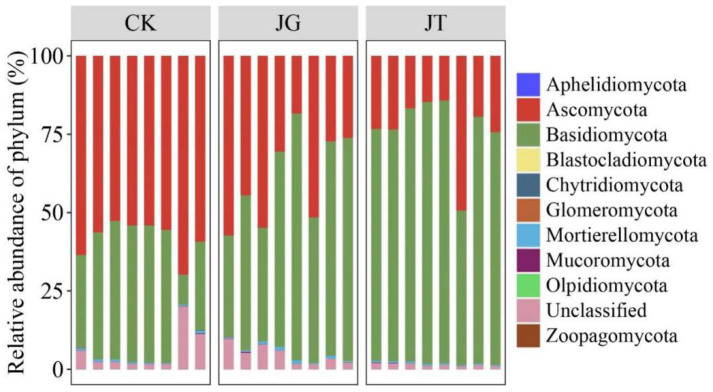
Fungi community composition relative abundance of different pots from the *L. hatsudake* orchard at the phylum level.

**Figure 3 microorganisms-11-02279-f003:**
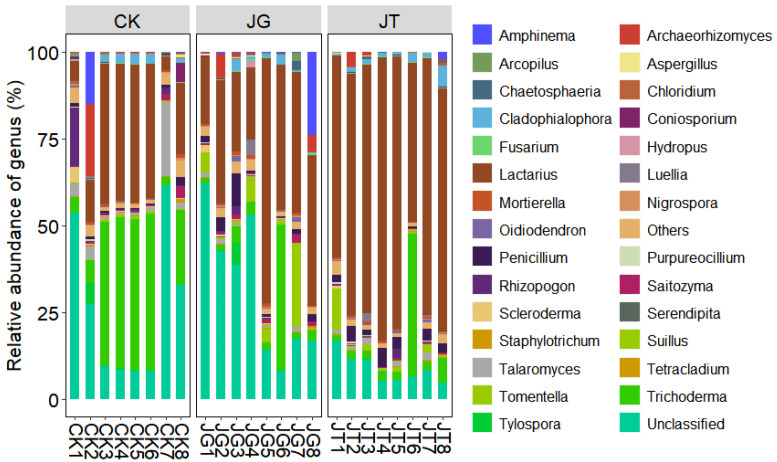
Fungi community composition relative abundance of different pots in the *L. hatsudake* orchard at the genus level.

**Figure 4 microorganisms-11-02279-f004:**
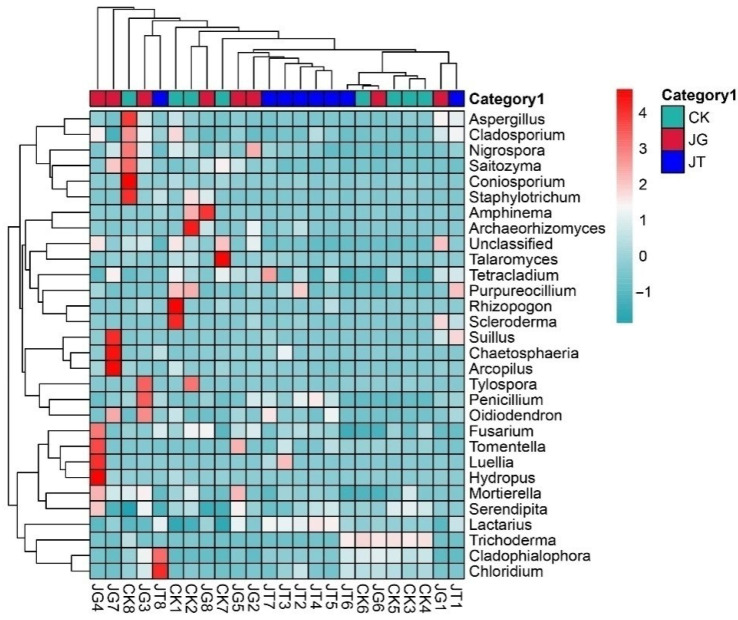
Heatmap of horizontal clustering analysis of fungi at genus level.

**Figure 5 microorganisms-11-02279-f005:**
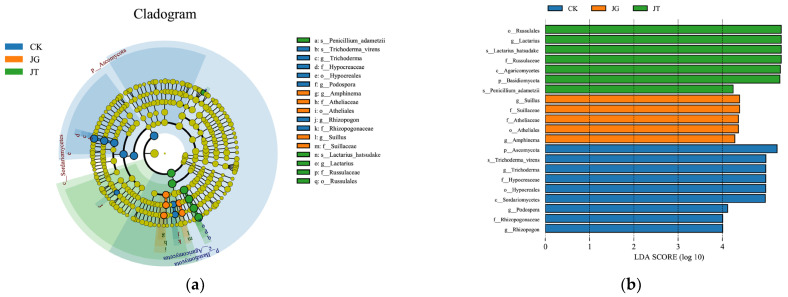
LEfSe analysis of fungi at genus level. (**a**) Evolutionary branching diagram of EfSe analysis. (**b**) Histogram of LDA value distribution.

**Figure 6 microorganisms-11-02279-f006:**
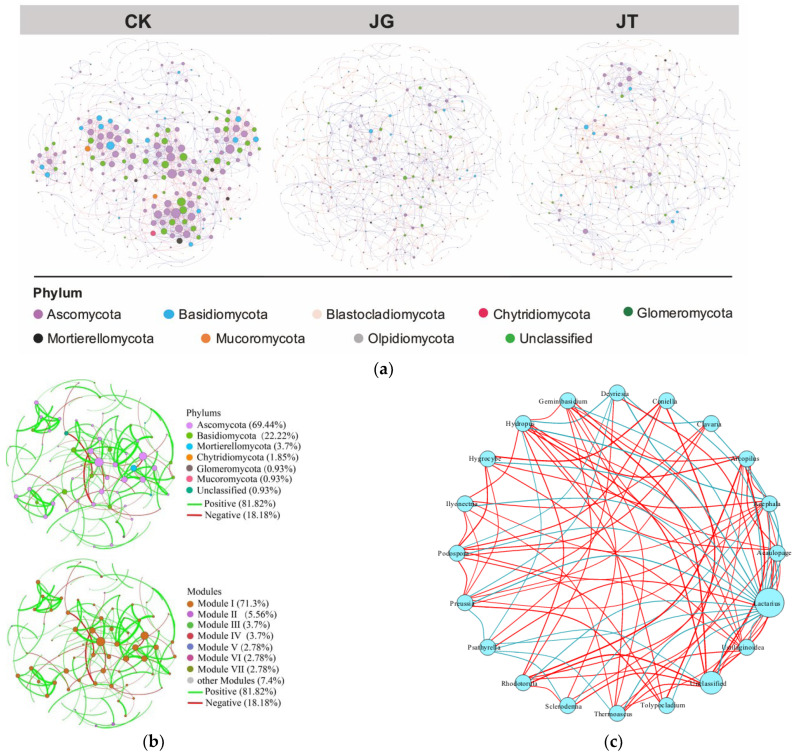
Interactions between the soil fungal community in the *L. hatsudake* plantation. (**a**) Horizontal collinear network interaction between the fungal communities at the phylum level during harvest. (**b**) Fungal network map. (**c**) Interaction network of the main fungi genera in the soil during harvest. The circle represents the species, and the size of the circle denotes the average species abundance. The lines represent the correlation between two species, and the thickness and color of the lines denote the strength and direction of the correlation, respectively.

**Figure 7 microorganisms-11-02279-f007:**
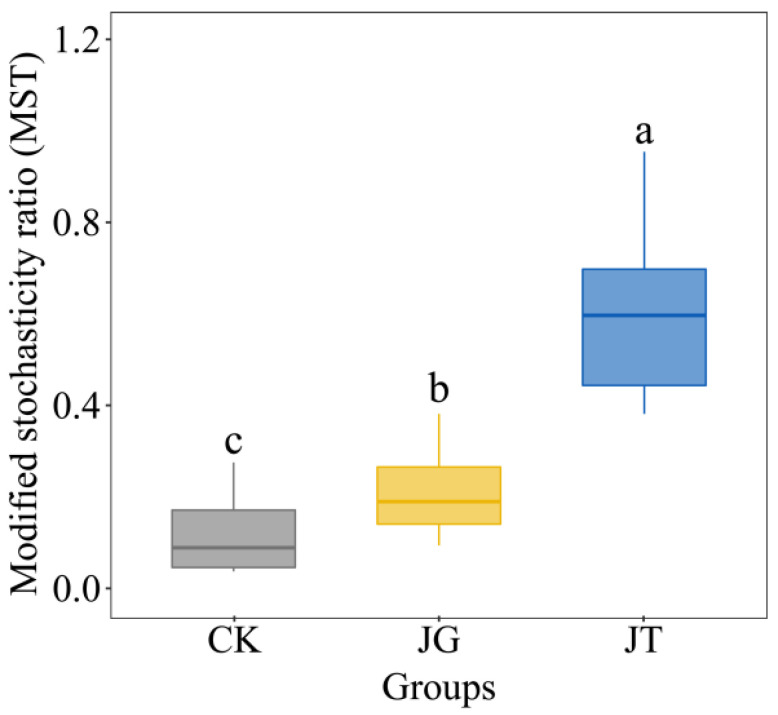
Modified stochasticity ratio. Note: The blue solid and dashed lines represent the best-fit value of the neutral community model and the 95% confidence interval of the model (estimated through 1000 bootstraps), respectively. OTUs with frequencies higher or lower than those predicted by the neutral community model are displayed in different colors. a, b, c: there is a significant difference.

**Table 1 microorganisms-11-02279-t001:** Alpha diversity of soil fungal communities of different pots from the *L. hatsudake* orchard.

Sample ID	Richness	ACE	Chao1	Simpson	Shannon
CK	511 ± 8.01a	595.88 ± 9.60a	616.87 ± 10.89a	0.78 ± 0.05a	3.77 ± 0.34a
JG	502 ± 8.90a	594.35 ± 10.16a	612.18 ± 13.62a	0.77 ± 0.05a	3.86 ± 0.44a
JT	474 ± 6.61b	569.76 ± 6.48b	573.05 ± 4.53b	0.49 ± 0.04b	2.42 ± 0.15b

Note: CK, non-mushroom-producing soil; JG, *L. hatsudake* mycorrhizosphere soil; JT, *L. hatsudake* fruiting body base soil. Data are represented as the mean ± standard deviation of eight samples; different lowercase letters within the same column indicate a significant difference between treatments (*p* < 0.05).

**Table 2 microorganisms-11-02279-t002:** Topological properties of soil fungal-community composition interaction networks in *L. hatsudake* orchard, including the empirical molecular ecological network and the randomized network.

Network	Index	Non-Mushroom- Producing Area	Mushroom-Producing Area	
		CK	JG	JT
Empirical	RMT threshold	0.910	0.890	0.890
	Total nodes	405	441	399
	Total edges	1360	647	586
	R^2^ of power-law	0.775	0.802	0.900
	Average degree (avgK)	6.716	2.934	2.937
	Average clustering coefficient (avgCC)	0.321	0.210	0.189
	Average path distance (GD)	5.592	9.850	9.260
	Positive edges	787 (57.9%)	308 (47.6%)	257 (43.9%)
	Negative edges	573 (42.1%)	339 (52.4%)	329 (56.1%)
Randomized	Average clustering coefficient (avgCC)	0.017 ± 0.002	0.007 ± 0.003	0.007 ± 0.004
	Average path distance (GD)	3.361 ± 0.008	5.569 ± 0.094	5.479 ± 0.090

## Data Availability

The data of this study are available from the correspondence author upon reasonable request.

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
