# Peer review of "Diversity and Network Relationship Construction of Soil Fungal Communities in Lactarius hatsudake Tanaka Orchard during Harvest"

_microorganisms, 2023, doi:10.3390/microorganisms11092279_

Round 1

Reviewer 1 Report

Lines 98-106: The authors need to write few words about the sampling plots , example the distance of the sampling plots (20X20m). They need to provide some information, very briefly about the soil of these plots such as the soil pH, moisture , and physical properties.(ex clay, sandy...).

Lines 123-141: It's a common problem when analysing DNA from soil samples, that the analysis indicates the presence of an organism at the present conditions or at the near past. I would like to see a comment on this in their document. 

The results are presented well. The article describes the  network relationships contraptions of the soil fungal communities, but the lack of any information regarding the soil conditions make the results and the conclusions related only to the present research.

Environmental information will expand any conclusions mined from the present work to a broader ecological interest.

Author Response

Thanks for your comments on our manuscript. We have revised our manuscript accordingly. Below are our point-to-point responses to the comments.

Point 1: Lines 98-106: The authors need to write few words about the sampling plots , example the distance of the sampling plots (20X20m). They need to provide some information, very briefly about the soil of these plots such as the soil pH, moisture , and physical properties.(ex clay, sandy...).

Response 1: Thank you for your suggestion. We have supplemented the sample distance information and soil information as follows, which are highlighted in red in the manuscript.

a) sample distance information

The distance between adjacent sampling points is about 2 meters.

b) soil information

The soil type of the plantation is red soil, with a pH value of 6.0-6.8 and a soil humidity of 45%-50% when collecting samples. (Line 108-110)

Point 2: Lines 123-141: It's a common problem when analysing DNA from soil samples, that the analysis indicates the presence of an organism at the present conditions or at the near past. I would like to see a comment on this in their document. 

Response 2: Thank you for your suggestion, we have taken into account this issue and following the method of Miae et al. [23] we strictly followed the sampling procedure. After sampling, the samples were quickly placed in an icebox and processed in a sterile laboratory environment. The samples were then promptly stored at -80 ℃ and transported to the sequencing company with full dry ice preservation. These measures ensured the maximum stability of soil microorganisms and consistency between the microorganisms during sequencing and sampling. (Line 111)

Reviewer 2 Report

The submitted manuscript is important due to the importance of Lactarius hatsudake Tanaka as an edible mushroom with rich economic and nutritional value.

There is main reason for a minor revision. In the methodology, the authors write that they used PCoA (line 160). The Authors named Fig. 1 as Principal component analysis (PCA). However, the description of the X and Y axes is again given as PCoA. If the authors used PCA, please write how they decided on the number of principal components. What was the eigenvalues of the correlation matrix and the relationship of the principal components and factor coordinates of the variables. What is indicated by the arrangement of variables in the graph relative to each other (on the same or opposite sides to each other). This information should be included in the methodology.

PCA is used to visualize the level of similarity among cases in a multidimensional data set. PCoA can be used to visualize distances between points by using f.e. Euclidean distance measures on the variables. PCA is used to identify common factors for primary variables. The Keiser criterion or the Cattell scree test is used to determine the number of principal components.

Line 11 - instead ..Tanakais … must be ..Tanaka is …

Author Response

Thanks for your comments on our manuscript. We have revised our manuscript accordingly. Below are our point-to-point responses to the comments.

Point 1: There is main reason for a minor revision. In the methodology, the authors write that they used PCoA (line 160). The Authors named Fig. 1 as Principal component analysis (PCA). However, the description of the X and Y axes is again given as PCoA. If the authors used PCA, please write how they decided on the number of principal components. What was the eigenvalues of the correlation matrix and the relationship of the principal components and factor coordinates of the variables. What is indicated by the arrangement of variables in the graph relative to each other (on the same or opposite sides to each other). This information should be included in the methodology.

PCA is used to visualize the level of similarity among cases in a multidimensional data set. PCoA can be used to visualize distances between points by using f.e. Euclidean distance measures on the variables. PCA is used to identify common factors for primary variables. The Keiser criterion or the Cattell scree test is used to determine the number of principal components.

Response 1: We are very sorry for the confusion, because of our carelessness, we named Figure 1 as Principal Component Analysis; in this article, we used Principal Coordinate Component Analysis, which we have corrected in the manuscript. (Line 228)

Point 2: Line 11 - instead ..Tanakais … must be ..Tanaka is …

Response 2: We apologize for our carelessness, we have corrected the error and marked it in red. (Line 11)

Round 2

Reviewer 1 Report

All the appropriate corrections have been made.